# Hydrologic resilience and Amazon productivity

Anders Ahlström [1,2], Josep G. Canadell [3], Guy Schurgers [4], Minchao Wu[2], Joseph A. Berry[5], Kaiyu Guan[6] & Robert B. Jackson[1,7]

The Amazon rainforest is disproportionately important for global carbon storage and biodiversity. The system couples the atmosphere and land, with moist forest that depends on convection to sustain gross primary productivity and growth. Earth system models that estimate future climate and vegetation show little agreement in Amazon simulations. Here we show that biases in internally generated climate, primarily precipitation, explain most of the uncertainty in Earth system model results; models, empirical data and theory converge when precipitation biases are accounted for. Gross primary productivity, above-ground biomass and tree cover align on a hydrological relationship with a breakpoint at ~2000 mm annual precipitation, where the system transitions between water and radiation limitation of evapotranspiration. The breakpoint appears to be fairly stable in the future, suggesting resilience of the Amazon to climate change. Changes in precipitation and land use are therefore more likely to govern biomass and vegetation structure in Amazonia.

[1] Department of Earth System Science, School of Earth, Energy and Environmental Sciences, Stanford University, 473 Via Ortega, Stanford, CA 94305, USA. [2] Department of Physical Geography and Ecosystem Science, Lund University, Sölvegatan 12, 223 62 Lund, Sweden. [3] Global Carbon Project, CSIRO Oceans and Atmosphere, GPO Box 3023, Canberra, ACT 2601, Australia. [4] Department of Geosciences and Natural Resource Management, University of Copenhagen, Øster Voldgade 10, 1350 Copenhagen, Denmark. [5] Department of Global Ecology, Carnegie Institution for Science, 260 Panama Street Stanford, Stanford, CA 94305, USA. [6] Department of Natural Resources and Environmental Sciences, University of Illinois at Urbana Champaign, W503, Turner Hall, 1102 South Goodwin, Urbana, IL 61801, USA. [7] Woods Institute for the Environment, and Precourt Institute for Energy, Stanford University, 473 Via Ortega, Stanford, CA 94305, USA. Correspondence and requests for materials should be addressed to A.A. (email: anders.ahlstrom@nateko.lu.se)

The future evolution of the global land sink of carbon represents one of the larger uncertainties for climate change projections. Altogether, tropical rainforests sequester more carbon than any other biome or land cover class globally[1, 2]. The Amazon basin alone accounts for ~40% of both the global tropical forest land area and above-ground biomass[3]. The resilience of the system to climate change depends on the mechanisms governing the forest–savanna transition. Generally, the transition follows a moisture gradient controlled by precipitation (P) and interacting with fire, but the transition can be abrupt between alternate stable states[4, 5]. Understanding the drivers and processes that govern the forest–savanna transition offers insights into the future resilience of the Amazon and other tropical forests[6–9]. It is not clear whether the Earth system models (ESMs) that are used to project climate and vegetation structure include the necessary processes to adequately explore the future of Amazonia (here defined as the Amazon basin and surrounding semi-arid vegetation).

Amazonia tightly couples the atmosphere and the land surface[10]. Global ESMs designed to account for feedbacks between vegetation and the atmosphere generally perform relatively poorly in the tropics when evaluated against carbon-cycle observation[11, 12]. This uncertainty originates from several issues, mainly terrestrial ecosystem sub-models, biases in climate generated internally in the ESMs, or both. The effect of climate biases on uncoupled carbon-cycle simulations has been shown to have a large impact on Amazonia[13]. Interpreting future model scenarios depends, in part, on understanding the basis of uncertainties and current biases.

Here, we combine empirical data and results from ESMs to evaluate large-scale ecosystem functioning in Amazonia and potential changes in the future. We first investigated the climate bias of the ESMs and their effect on simulated water and carbon fluxes, and vegetation structure. We find that most differences among models are related to their internally simulated climate, particularly P. Using this information, we designed an analysis to account for climate biases by transforming the spatial information from the ESMs to P-based relationships that represent the hydrologic and ecosystem dependency on P.

We focused on the forest–savanna transition using outputs from ESMs and independent empirical data. We calculated dependencies on P in Amazonia using information on evapotranspiration (ET), gross primary productivity (GPP), above-ground biomass (AGB) and tree cover by aggregating these indicators at intervals of P over the Amazon basin and surrounding semi-arid regions. The resulting functional relationships are especially valuable when analysing empirical data and ESM outputs because they illustrate the response to P and are therefore less sensitive to biases in the ESMs. The functional relationships that we found in empirical data share three main features: an initial steady increase of ET, GPP, AGB and tree cover with increasing P, a breakpoint at ~2000 mm annual P, after which the indicators do not increase with P, and a significant deviation from the steady increase trend between 1200 and 2000 mm annual P, showing a large and abrupt change in ET, GPP, AGB and tree cover at the forest–savanna transition. We next explain the main differences between the empirical and simulated relationships, showing that models and data agree fairly well on the functioning and drivers of observed physical and ecological changes along the forest–savanna transition. The underlying ecohydrological explanation for the relationship to P that is shared between vegetation structure and water, and carbon fluxes offers insights into the future resilience of Amazonia and, more broadly, of tropical forests, and what factors may cause the moist forest to shift to a lower biomass state.

## Results

**Climate model bias.** Combining internally generated climate and GPP from an ensemble of nine CMIP5[14] ESMs over Amazonia reveal large climate biases (Fig. 1). On average, the models show a negative P bias of −412 mm $yr^{-1}$ (Fig. 1a). This dry-bias coincides with low cloudiness and a general overestimation of shortwave radiation affecting GPP (Fig. 1b, c). Altogether, in a linear regression, the three main climatic drivers P, shortwave radiation and temperature explain 93% ($R^2 = 0.93$) of the differences in simulated GPP. The overall dry-bias also leads to a general underestimation of the extent of Amazon rainforest and the location of the forest–savanna transition (Fig. 2a–k) as well as biased predictions of both total AGB (Fig. 2l) and the biomass that can be lost or gained under future climate change, compromising the initial conditions of future simulations of Amazonia.

**The forest–savanna transition.** To control for climate biases in mean P (as the climate variable hypothesised to lead to the largest

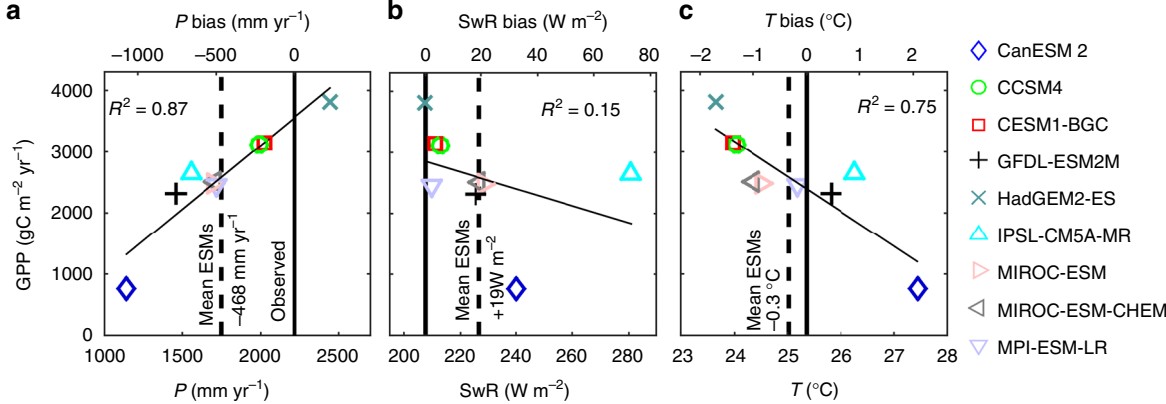

**Fig. 1** Earth system model climate drivers and gross primary productivity averaged over the Amazon basin for 1981–2005. **a** Mean Amazon basin Earth system model (ESM) gross primary productivity (GPP) dependency on ESM mean annual precipitation. *Black* represents observation-based reference Amazon basin mean annual precipitation[56, 57] (P), with ESM biases on upper X-axis. **b** ESM GPP dependency on ESM mean annual downward shortwave radiation (SwR) and bias relative to shortwave radiation from CRUNCEP[58]. **c** ESM GPP dependency on ESM mean annual temperature (T) and bias relative to temperature from CRUNCEP. In a multiple regression, the three climatic drivers together explain 93% ($R^2 = 0.93$) of the differences in simulated GPP between the ESMs

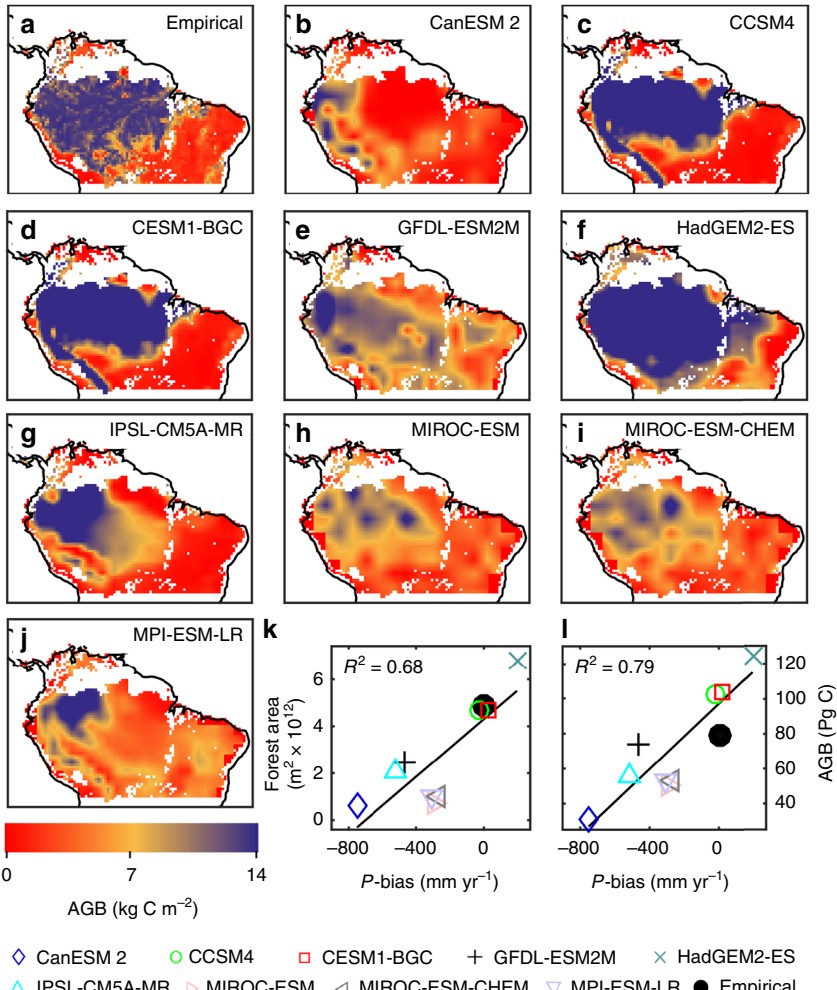

**Fig. 2** Spatial above ground biomass distributions. **a** Empirical above ground biomass (AGB). **b–j** Earth system model (ESM) AGB. **k** Relationship between ESM-simulated tropical forest area and precipitation (*P*) bias over the Amazon basin and surrounding semi-arid regions. The forest area was defined as the sum of the area of all grid cells with AGB > 10 kg C m$^{-2}$ in maps **b–j**. *Symbols* follow legend of Fig. 1 and the empirical data set is shown as a *solid black circle*. **l** Relationship between AGB sum and P bias

biases in ecosystem properties), we transformed spatial information from empirical data sources and ESMs into P-based relationships over the Amazon basin and surrounding semi-arid regions by aggregating ET, GPP and AGB at P intervals. In this way, we can compare the functional responses of the simulated systems to P, uninfluenced by biases in mean Amazon basin P. In addition to the nine ESMs introduced above, we included ET and GPP empirically upscaled from global flux tower measurements[15], two independent diagnostic estimates of ET[16–18], a remote sensing based estimate of tree cover[19] and a satellite-derived vegetation optical depth (VOD)-based estimate of AGB[20] over the Amazon basin and surrounding semi-arid regions (Fig. 2; Supplementary Fig. 1).

ET from empirical data sets and ET from the ESM ensemble show a similar response to P; ET initially increases with P until ~2000 mm yr$^{-1}$ after which ET is fairly stable and does not increase with additional P (Fig. 3a). The general shape and breakpoint of P at ~2000 mm yr$^{-1}$ are also found in the response of GPP and AGB to P (Fig. 3b, c), making it the dominant feature determining hydrology, ecosystem productivity and vegetation state. Previous studies have identified a similar breakpoint for the emergence of tropical forests and for tree cover in general, also at ~2000 mm annual P[4, 5]. The overall shape of the curve is reminiscent of a well-known hydrological relationship, describing the two regimes of ET

control: a water-limited regime and an energy-limited (radiation) regime. A separate empirical analysis[21] of the current hydrologic cycle in tropical forests globally comes to a very similar breakpoint of 2000 mm yr$^{-1}$ (range: 1850–2150 mm yr$^{-1}$) for the transition from water-limited to radiation-limited (water-saturated) ET and vegetation productivity. The average breakpoint determined here using a trend break analysis of the empirical data sets dependency on P is 2023 mm yr$^{-1}$ with a standard deviation of 128 mm yr$^{-1}$ (dashed red lines in Fig. 3, breakpoint tree cover: 2075 mm yr$^{-1}$, Supplementary Fig. 2).

The breakpoint we determined occurs where annual P is sufficient to recharge water stores to sustain dry-season transpiration and productivity. The breakpoint is therefore associated with the annual balance of P and potential ET (PET) —the land-surface supply and the atmospheric demand for water —but it is also related to the intensity and seasonality of P and water gains or losses through lateral flow (run on/off). Above the breakpoint, where water is readily available, GPP and vegetation growth are increasingly limited by sunlight as clouds reduce incoming shortwave radiation[22]. Empirical data and the ESM ensemble mean generally agree on the ET, GPP and AGB response to P, but the empirical data show a sharper decrease when transitioning from the radiation-limited regime to lower P levels (Fig. 3). This sharper decline for the empirical data suggests

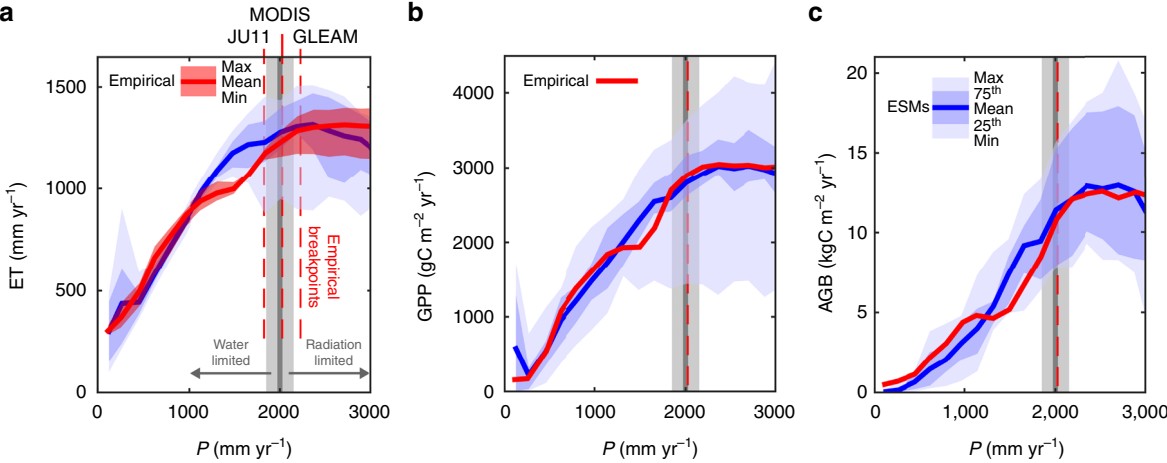

**Fig. 3** Functional precipitation relationships. **a** Evapotranspiration (ET) response to precipitation (*P*) from nine ESMs (*blue shaded areas* and *solid line*) and calculated from three empirical data sets, MODIS MOD16[16], upscaled eddy-flux tower estimates[15] (JU11) and GLEAM v3.0a[17, 18]. *Grey vertical line* and *shading* represent an independent, empirical, estimate of the breakpoint between water-limited and radiation-limited control of GPP[21] (gross primary productivity). The *shading* represents the range of estimated breakpoints across tropical forest locations globally. *Dashed red lines* show breakpoints in the empirical data sets, found in a trend break analysis (Supplementary Fig. 2). **b** GPP response to P of ESMs and empirical GPP from upscaled eddy-flux estimates[15]. **c** AGB response to P of ESMs and satellite-derived vegetation optical depth (VOD)-based estimates of AGB[3]. All empirical data products were combined with P from GPCC v7[57] corrected following[56]

the presence of a tipping point, where the moist highly productive and carbon-dense forest rapidly transitions to a lower biomass state. The sharp decline in AGB resembles the previously observed bimodal distribution at the forest–savanna transition[5], which has been suggested to represent alternative stable states[4]. Understanding the underlying cause for this missing tipping point in the ESMs is pivotal for characterizing the system's resilience to a potential future drying of the Amazon.

**The missing tipping point**. The observed sharp transition to a lower biomass state may be driven by fires[5]. Vegetation processes such as mortality[23] and fires[24] are poorly captured in most ESMs[25]. However, land use and forest clearing result in similar decreases in ET, GPP and AGB; it is unclear to what extent they contribute to the model-data discrepancies identified here. To investigate this issue, we combined ET and the vegetation variables with P and information on land use[26].

Above ground biomass in the Amazon basin and surrounding semi-arid ecosystems shows a bimodal distribution, which causes the sharp transition (Fig. 4a). The frequency field that combines P, AGB and land use indicates two dominant states: a high biomass and low land use state, with P above the ~2000 mm breakpoint representing relatively pristine tropical forest, and a lower biomass, high land use state centred at $P \sim 1500$ mm, suggesting that land use may drive the apparent tipping point. To evaluate the influence of land use on the functional relationship between P and vegetation properties, we increasingly excluded grid cells based on their fraction of land use and recalculated the mean relationship. We found that the sharp decline of AGB with P below the breakpoint decreases when grid cells with high proportions of land use are removed. Furthermore, the sharp decline almost disappears when only grid cells with <20% of their area under land use are included.

We next evaluated the role of fire using a satellite-based estimate of burned area fraction[27, 28] (Fig. 4b). Evaluating all grid cells, the burned area fraction peaks in the upper half of the water-limited regime, between 1250 and 2000 mm yr$^{-1}$, coinciding with the P interval where models and data diverge (Fig. 3). Removing locations under land use from the analysis shifts the

peak of the distribution towards lower P levels, suggesting that land use decreases fires in drier regions and increases fires at the savanna-forest boundary. This shift due to land use would be consistent with active suppression of fires in dry regions and the use of fires to clear forest at the land use expansion frontier.

Similar effects of land use were found on ET, GPP and tree cover[19] (Fig. 4c–e). Although ET and GPP change less in heavily managed locations than AGB and tree cover do, all empirical data sets suggest that the mean response to P and the transition between ecosystem states is smooth when removing land use effects. The shift from a sharp decline to a smooth transition between ecosystem states when accounting for land use suggests that there is little risk for a general ecosystem state tipping point. Instead, the sharp transition is mainly caused by human transformation of highly productive ecosystems close to the breakpoint as well as widespread land use in systems with ~1500 mm annual P, which decrease the productivity and biomass in the semi-arid Cerrado. Our analysis does not exclude the existence of reported fire-driven bi-modality[5], which leads to separate and stable vegetation states[4]. Such bi-modality, with biomass at two distinct states for similar P does not necessarily cause a tipping point in our analysis of functional relationships, which represent the average biomass in a given P interval. Overall, however, the deviation from the hydrological constraint observed here appears, currently, to be related primarily to land use.

The ESM simulations presented here were forced with the same land use information[26] used in the empirical analysis of land use effects presented above. ESMs therefore accounted for the land use and land use change by prescribing land clearing, pastures and croplands, but their ensemble mean P dependency does not show the decrease that is caused by land use (Fig. 3). The reason that ESMs do not correctly capture the effects of land use is their considerable biases in the spatial distribution of P (Fig. 5). These spatial P biases lead to a biased relationship between P and land use (Supplementary Figs. 4 and 5), which results in the incorrect replacement of carbon-dense moist forest with managed agricultural or grazing lands in most ESMs. All ESMs except for HadGEM2-ES inaccurately predict P at levels associated with semi-arid savannas over large parts of central Amazon (Fig. 5). HadGEM2-ES in turn, as the sole ESM that predicts realistic P, is

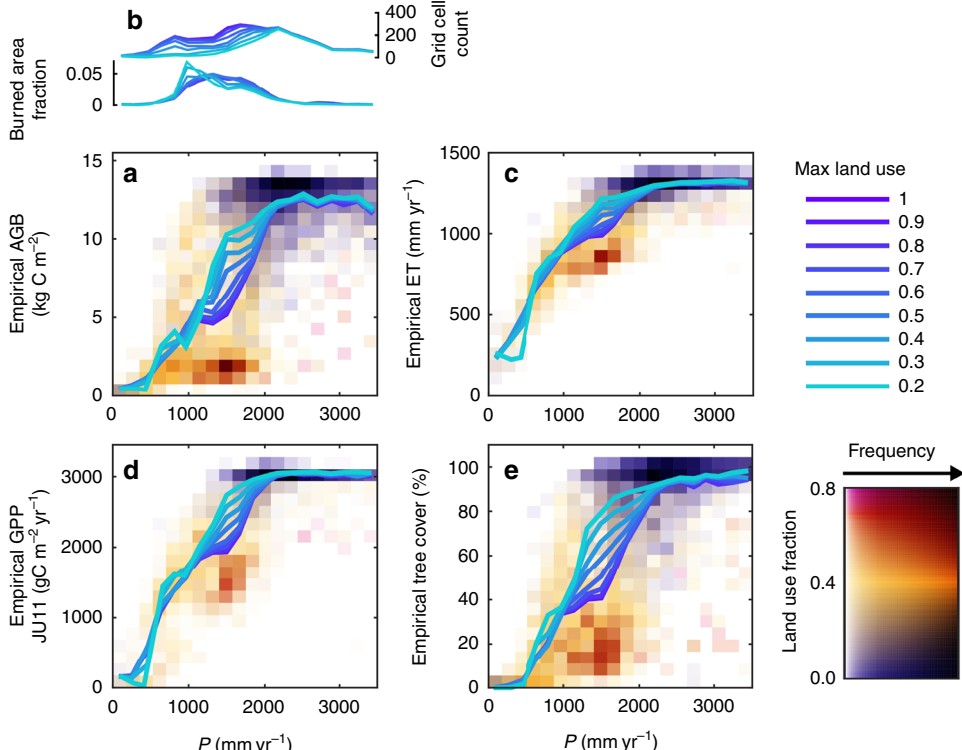

**Fig. 4** Functional relationships and their dependency on land use. **a** Dependency of above ground biomass (AGB) on precipitation (*P*). *Coloured frequency field* shows the dependency of AGB on P and land use. *Colour* follows the mean land use fraction in a P-AGB bin, and the saturation describes relative frequency. *Coloured lines* represent the functional relationships, found when averaging AGB over P-bins. The *colours* describe the maximum land use fraction of the grid cells included in the calculations of the functional relationships; from all grid cells (1) to only include grid cells with land use fraction lower than 0.2. This procedure was repeated for burned area fraction which is shown in **b** together with the number of grid cells available per land use limit. **c** through **e** show the corresponding frequency fields and functional relationships of evapotranspiration (ET) **c**, gross primary productivity (GPP) **d** and tree cover **e**. For ET **c** the spatial distribution of mean ET and P were calculated from the three data sets before the analysis of the functional relationships, where P differ only in the time period used for the climatology (individual *panels* for the three ET data sets are shown in Supplementary Fig. 3)

the only model to accurately capture the effect of land use (Supplementary Fig. 3). This general insufficiency to provide climate that allows moist tropical forest differ between the ESMs. Two ESMs (CanESM2 and GFDL-ESM2M) incorrectly predicted semi-arid climate over the entire Amazon basin. Another two ESMs (CCSM4 and CESM-BGC) predict a reasonable areal coverage of water-saturated climate associated with moist forest but simulated incorrect spatial distribution, which extended into the agricultural zone and leads to incorrect clearing of forest. Therefore, the ESMs generally predict tropical forest extent and Amazonian biomass that are too low compared to observations (Fig. 2). Furthermore, due to their spatial P biases, most ESMs do not accurately capture the influence of land use in their functional P relationships, which explains the disagreement between the ESMs and empirical data.

**Future changes**. Future climate change is predicted to increase PET[29] and intensify the seasonality of P[30], both of which could shift the hydrological breakpoint to wetter levels of P. However, the ESMs considered here do not show a marked shift in their ET dependency on P even under RCP 8.5 (Fig. 6a), suggesting that the potential effect of climate change on the breakpoint may be relatively small. Increased $CO_2$ concentrations are predicted to increase water use efficiency (WUE), shifting GPP upwards, but the location of the breakpoint between the regimes is not directly affected by WUE in these simulations (Fig. 6b). The ESM ensemble predicts an increase in ecosystem water use efficiency (eWUE; GPP/ET) of ~30% between 1981–2005 and 2081–2100 (Supplementary Fig. 6). This increase in eWUE originates

primarily from an increase in plant WUE (GPP/transpiration), which is induced by $CO_2$ fertilisation and/or effects on canopy conductance (Supplementary Figs. 6–8).

The WUE increase indicates relatively large GPP resilience to drying by maintaining current GPP even under large reductions in P (Fig. 6b). The effect of WUE on GPP is relatively constant over the entire P range, which is illustrated by the good agreement between the simulated future GPP response in ESMs to P and the concurrent empirical GPP response to P when scaling it with the ESMs WUE increase (Fig. 6b). Although the ESMs predict a future increase in GPP for all P levels, predicted increases in AGB are isolated to a narrow P range above the breakpoint (Fig. 6c). In contrast to GPP, AGB does not increase in the water-limited regime. This difference in WUE-induced increases decouples growth from productivity changes, echoing empirical studies reporting increased WUE but not growth[31, 32].

## Discussion

We have shown that ESMs can accurately predict Amazon forest structure and functioning after their biases in internally generated climate are corrected. The ecohydrologic constraint is found in all data sets and in the global fully coupled ESMs. This important ecohydrological relationship and its breakpoint provide insights into how Amazonia may respond to climate change. Moreover, biases in internally generated climate explain most of the differences between ESM predictions of water and carbon fluxes, and ecosystem state and structure. The general dry-bias of ESMs compared with observations implies that most ESMs predict too

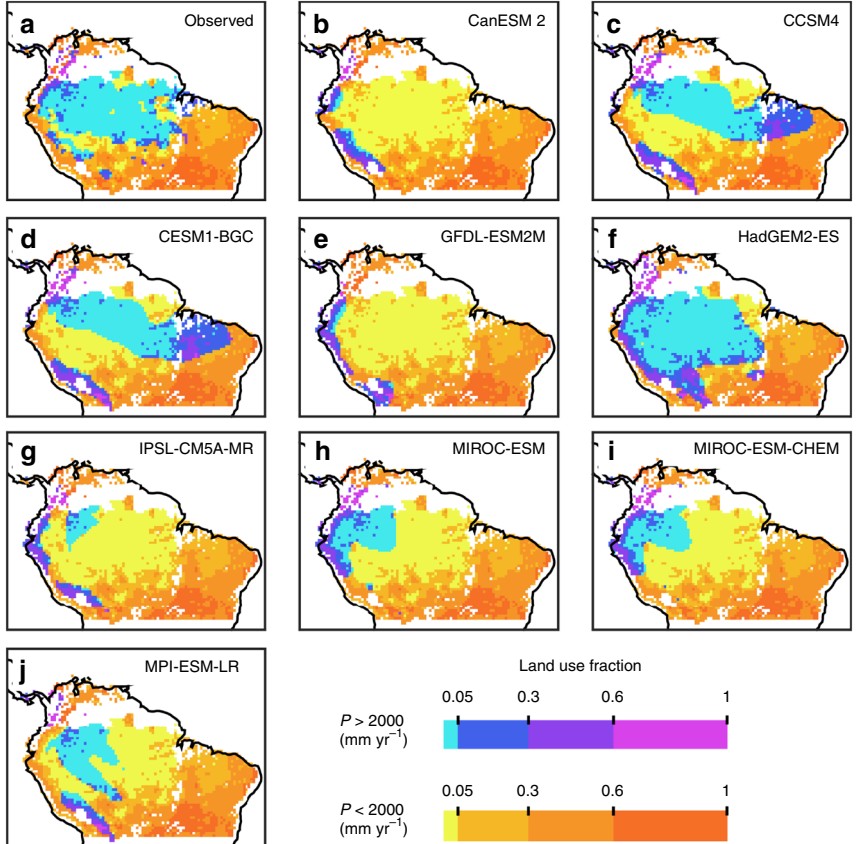

**Fig. 5** Spatial distributions of land use and water-limited and radiation-limited precipitation for 1981–2005. **a** Observed precipitation (*P*)[56, 57] and land use[26]. P was divided into water-limited (*P* < 2000 mm yr⁻¹) and radiation-limited (*P* > 2000 mm yr⁻¹) categories based on the empirical 2000 mm annual P regime breakpoint. Land use fractions was divided into four classes and used to vary the *colours* (*yellow* to *red* and *blue* to *magenta* for water-limited and radiation-limited, respectively). **b–j** The corresponding P-land use maps for each of the nine Earth system models (ESMs)

much of the Amazon to be water-limited, explaining previously reported uncertainties and lack of agreement with empirical data[11, 12]. Recent studies highlights ESM difficulties in simulating atmospheric convergence, convection[33] and low-level cloud cover[34], which likely explain at least part of the dry-bias shown here. The relatively consistent response of ET to P across the ensemble suggests that most ESMs would provide reasonable land-surface feedbacks if provided with unbiased climate. Although we have not identified the underlying cause of biases in the ESM climate predictions, our findings suggest that under-estimated annual ET from the land-surface is not the dominant cause of the climate biases.

The empirical data sets used here have their own uncertainties. Uncertainties in individual data sets are difficult to assess because they originate in part from sparse observations (e.g. flux towers[35] or weather stations[36, 37]) and from clouds interfering with satellite observations in the wet season[38]. Our approach to address the uncertainty includes using multiple independent empirical data sets that represent both ecosystem fluxes and states. All data sets show a similar breakpoint between water and radiation limitation (Supplementary Fig. 2). This breakpoint, apparent from the functional relationships between different ecosystem states and fluxes, and precipitation is also similar to the breakpoint found previously using interannual variations in ecosystem productivity and a gravity informed water balance model[21]. Altogether, the coherence in the existence and location of the breakpoint across ET, GPP, AGB and tree cover and previous work lends support to the existence of a breakpoint with a value close to 2000 mm annual P.

Our assessment of future changes in Amazonian ecosystems is described as a response to P and does not assess the mean ecosystem response to changing climate. We thereby control for the large uncertainty between simulated changes in climate, especially P (Supplementary Fig. 9), which is a large cause for reported divergent future predictions of Amazonian ecosystem fluxes and structure[39, 40]. However, the ability of ESMs to accurately simulate Amazon responses to future climate may be compromised by relatively poor representation of drought mortality and disturbances[25, 41], leading to large uncertainties in vegetation turnover[42] and resulting biomass. Uncertainties in the models' ability to capture the occurrence and the ecosystem response to potentially more extreme weather and hotter droughts may also affect the future responses to P presented here. This potential impact of increased drought mortality remains to be evaluated and requires a new generation of ESMs with land models that include detailed representation of vegetation dynamics, disturbances and mortality, with accurate predictions of future climate and weather extremes.

Future potential expansion of the water-limited regime under climate change poses risks for the carbon storage and biodiversity of the moist regions of Amazonia. The state of art ESMs used here account for feedbacks between the atmosphere and land surface under climate change and do not simulate any marked changes to the value of the ecohydrological breakpoint, suggesting resilience of the Amazon forest to climate change. However, a potential upward shift of the hydrological breakpoint would have a similar effect as a decrease in mean P, increasing the extent of the water-limited regime. Given the importance of the breakpoint and the relatively crude resolution of the global ESMs presented here, we

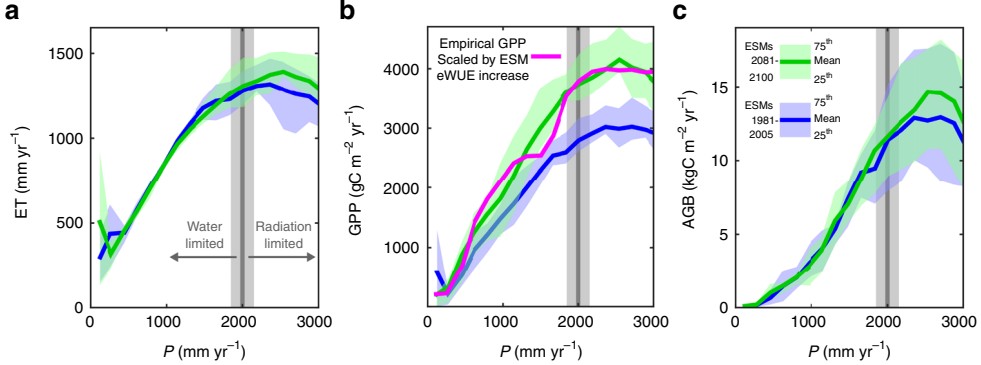

**Fig. 6** Future changes in evapotranspiration, gross primary productivity and above ground biomass under RCP 8.5. **a** Earth system model (ESM) mean and inter-quartile range of historical (*blue*) and future (*green*) evapotranspiration (ET) dependency on precipitation (*P*) The *grey line* and *shading* represent the mean and range, respectively, of estimated breakpoints across tropical forest locations globally[21]. **b** Historic and future gross primary productivity (GPP) dependency on P, *coloured* for ESMs as in **a**. The *magenta line* represent the empirical GPP dependency on P scaled with the ESM ensemble increase in ecosystem water use efficiency (eWUE; 1.31). **c** Historic and future dependency of above ground biomass (AGB) on P. The eWUE-induced increases seen for GPP do not translate to similar increases in AGB, and are limited to a narrow P range in the radiation-limited regime. In the water-limited regime, AGB does not change, indicating that $CO_2$ effects on eWUE do not affect AGB in the water-limited regime in the ESMs

suggest that future more detailed studies need to be performed to confirm the stability of the breakpoint.

Besides a shift in the breakpoint, changes in either the mean or variability of P could shift the Amazon towards the water-limited regime. These changes could occur in three ways. First, a decrease in mean annual P could push the entire Amazon or sub-regions towards the water-limited regime. However, evaluating simulated Amazon P change from a larger ensemble of 21 ESMs and general circulation models (Supplementary Table 1) under RCP 8.5 shows no agreement for such a transition (Supplementary Fig. 9). About half of the models (12/21) predict a drying, but there is no apparent relationship between their current performance and their predictions of future P, making an assessment of the likelihood of drying difficult. Second, even with no change in long-term mean P, increased variability or stronger extremes[43] imply larger or more common departures into the water-limit regime. In addition, because the system responds more strongly to a decrease in P than to an increase, the response to increased interannual variations is asymmetrical and would likely decrease mean GPP[44]. Third, a recent statistical analysis links the current performance of climate models to a future strengthening of the dry season[30]. Years of large fires have been found to co-occur with drought and water-limited conditions[45, 46], and a longer dry season increases the time spent in the water-limited regime where fires can occur. The impact of a more prominent water-limited regime due to one or more of these factors could be amplified by WUE-induced increases in GPP, which would provide more flammable material for fires and further contribute to a shift to the lower biomass state.

Our results reconcile models and data, and highlight the dominant role of the eco-hydrologic constraint in Amazonia. We found that climate biases, especially in precipitation, limit the current ability of individual models to represent the Amazon and likely other tropical forests accurately. Interactions between precipitation biases and land use are another cause of discrepancies among models and between models and empirical data sets. Although future $CO_2$ effects on WUE will likely increase GPP, increased climate variability can reduce carbon accumulation in Amazonia. The hydrological regime and its relationship to the breakpoint identified here may ultimately govern Amazon biomass and vegetation structure of native ecosystems.

## Methods

**Earth system models and their results**. Outputs from nine ESMs contributing to the Coupled Model Intercomparison Project phase 5 (CMIP5)[14] were analysed (Supplementary Table 1). We used $CO_2$ concentration-forced historical realisations and future projections following Representative Concentration Pathway 8.5 (RCP 8.5) $CO_2$ concentration[47] and land use change scenarios[26]. All ESM outputs were interpolated to a common $0.5 \times 0.5°$ grid using nearest neighbour interpolation. In addition to the ESMs used in the full analysis, precipitation outputs from an additional 12 climate models were used to investigate future precipitation changes (Supplementary Table 1). The majority of these additional 12 models did not include representations of dynamic vegetation and carbon cycle. For both GPP and AGB, we analysed the time period 1982 through 2005 for the historical period and 2081 through 2100 for the future period.

To facilitate comparison of above ground carbon with the empirical data set, we removed the reported root carbon from total vegetation carbon. For two ESMs (GFDL-ESM2M and MPI-ESM-LR), root carbon was not available. For these two ESMs, we estimated root carbon based on the remaining models' root-to-total vegetation fraction (0.1566). This fraction was calculated over the entire region of analysis.

**Empirical data products**. Empirical estimates of GPP and one of the ET products originate from upscaled FLUXNET eddy-covariance tower measurements[15]. The overall upscaling procedure involves three main steps: (I) processing and quality control of the FLUXNET data[48, 49] and aggregating the half-hourly flux data to monthly means. (II) Training a machine-learning-based regression algorithm (Model Tree Ensembles, MTEs[50]) for tower observations using site-level explanatory variables and satellite observed fraction of absorbed photosynthetic active radiation (fAPAR). Explanatory variables (n = 29) include fAPAR from the SeaWiFS sensor, climate variables, vegetation type from the IGBP classification and information on photosynthetic pathway. (III) Applying the established MTEs for global upscaling, using gridded data sets of the same explanatory variables. Individual model trees (n = 25) were forced for each biosphere-atmosphere flux using gridded monthly inputs from 1982 through 2011. The median over the 25 estimates for each pixel and month is used as the best estimate of a biosphere-atmosphere flux for further analysis.

In addition to the ET from upscaled flux tower measurements, we used MODIS ET[16] and GLEAM v3.0a[17, 18] as additional, independent estimates of ET. MODIS ET combines a modified Penman–Monteith equation[51] with satellite-based remote sensing data and climate information at 5 min spatial resolution (1/12°). GLEAM v3.0a combines satellite-based remote sensing with the Priestley–Taylor equation[52], the Gash[53] analytical interception model and climate data at 15 min spatial resolution (1/4°).

The AGB data set[3] originates from satellite-based vegetation optical depth data[20]. Information on VOD was linked to AGB by an empirical relationship between VOD and tropical AGB[54]. Using the empirical relationship, VOD was translated to AGB globally. We used the full available period, 1993 through 2012, using precipitation representing the same years in the comparison between empirical AGB and P. The tree cover data set origins from Landsat satellite remote sensing data with 30 m spatial resolution[19]. Trees were defined as trees that are taller than 5 m.

We used burned area from the Global Fire Emissions Database version 4 (GFED4s), which is based on an updated version of ref. [27] with burned area from ref. [28] and small fire burned area from ref. [55], summing the reported monthly burned area fraction to annual amounts. The annual time series were then averaged over the time period 1997 through 2014 and compared against precipitation averaged over the same time period.

**Data harmonisation and geographical boundaries**. All data were resampled to a common grid with a resolution of 0.5° × 0.5° in longitude and latitude. The analysis was performed over the Amazon hydrological basin as well as over the surrounding semi-arid regions as defined by ref. [1] between latitudes 16° N and 20° S over South America (Supplementary Fig. 1).

**Data availability**. The data that support the findings of this study are all publicly available from their sources. Processed data, products and code produced in this study are available from the corresponding author upon reasonable request.

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

## Acknowledgements

A.A. acknowledges support from The Royal Physiographic Society in Lund (Birgit and Hellmuth Hertz' Foundation) and the Swedish Research Council (637-2014-6895). J.G.C. thanks the support of the National Environmental Science Programme ESCC Hub and R.B.J. acknowledges support of USDA/NIFA Grant #2012-68002-19795. We acknowledge the World Climate Research Programme's Working Group on Coupled Modelling, which is responsible for CMIP, and we thank the climate modelling groups listed in Supplementary Table 1 of this paper for producing and making available their model output. For CMIP, the US Department of Energy's Program for Climate Model Diagnosis and Intercomparison provides coordinating support and led development of software infrastructure in partnership with the Global Organization for Earth System Science Portals. We thank the developers, providers and contributors to the empirical data sets used in this paper; the Jena MTE GPP product, GLEAMv3.0, MODIS ET, GFED4s, GPCCv7, CRUNCEP, LUHa.v1, the tree cover and AGB data sets. This study is a contribution to the strategic research area Biodiversity and Ecosystem services in a Changing Climate (BECC). It also contributes to the efforts of the Global Carbon Project to understand perturbations to the carbon budget and its future dynamics.

## Author contributions

A.A. led the writing, design of the paper and analysis in discussion with all authors. All authors contributed to the text.

## Additional information

**Competing interests :** The authors declare no competing financial interests.

