## [Peer Review File · Nature Communications]

Reviewers' Comments:

Reviewer #1 (Remarks to the Author)

The manuscript the relationship between long-term GPP, ET and AGB to long-term precipitation (P), using an ensemble of nine CMIP5 ESMs, and a couple of observed datasets. They found that the above quantities have a linear relationship to P if $P < 2000$ mm/yr, being stable (or radiation-limited) if $P > 2000$ mm/yr. The authors go on and investigate these relationships using observed data, but filtering for different levels of land use, and conclude that these relationships become steeper if land use is higher. Their final conclusion is that reducing land use change emerges as a key opportunity to reduce vulnerability of Amazonian ecosystems.

I overall agree with the authors reasoning and conclusions. I just think their results are not new, and we have known this since the 2000s, may be since the 1990s. All these arguments have been repeatedly used by authors that discussed the Amazon tipping point. For example, the argument that deforestation would decrease the resilience of the Amazon rainforest was first presented by Shukla et al. (Science, 1990).

The main contribution of the authors may be to demonstrate that the relationships GPP, ET, AGB hold for the ensemble of the nine CMIP5 models they picked. However, the lack of predictability for future scenarios ("inconclusive support for such a transition") makes the results less interesting. Again, even the inconclusiveness is not new, having first been reported by the first edition of C4MIP in 2006.

The way data is presented in Figure 4 is also new and interesting, but again, there is not any new information in it. It is well known that deforestation (increased land use) decreases AGB (there is nothing more obvious than that), ET, and GPP, and in the way they plotted their figures, these changes happen in the range of precipitation where land use occurs.

My recommendation to the authors is to publish the analyses around Figures 1-3 in a more specialized journal, with a remark, however. The authors used only global datasets in their study, having cropped the Amazon region for their analyses. Global datasets can have significant regional biases, and uncertainties in empirical, or observed datasets, just cannot be avoided. I recommend using more regional studies in their analyses (they have cited very few regional studies in the specialized literature), and using an ensemble of observations in addition to their ensemble of models.

Reviewer #2 (Remarks to the Author)

One major set of related comments:

Line 160-169 – The reasoning here is very unclear. What does this sentence even mean: "Such bimodality that overlaps on the P-axis does not necessarily cause a tipping point"? This passage is very poorly written.

Additionally, the reasoning overreaches substantially. It is quite clear and well known that the cerrado and other Amazonian grassy systems are used much more intensively than Amazonian forests. Thus, land use differs between forest and savannas. However, land use does not cause the cerrado to be where it is – after all, the cerrado clearly pre-dates colonial land use practices. So the directionality of the causality that is inferred here is totally wrong.

Line 260-262 – This is a totally unsupported conclusion. Moreover, it's not even necessary. The other key conclusions are worthwhile and interesting. It's also quite clear that, today, fires within

forest have anthropogenic origins related to land use change – so you can make the same exact conclusion on line 264 without being unnecessarily combative about fire-vegetation feedbacks.

So, if you remove Lines 160-169 and Lines 260-262, the paper is interesting and makes real and worthwhile contributions to the literature on Amazon forest susceptibility to global change.

Some minor comments:

You should introduce fire more completely in the introduction, given how fundamental it is to your later analyses. You could also or alternatively reframe a bit to make fire a bit less central, which you should probably do anyways, as described above.

Line 44 – P = phosphorus, and abbreviating like this in the introduction is unnecessary/confusing.

Line 97 – should be 'relationships'

Line 179 – most geographers would describe the central Amazon as being in the region of Manaus. There are no semi-arid savannas anywhere near there. Do you mean central South America? Central Brazil? And actually, a good part of your study region is not actually in Amazonia.

Reviewer #3 (Remarks to the Author)

General comments

In this study, the authors investigate the problem of the Amazon forest vulnerability to climate change by using a subset of CMIP5 earth system models (ESM) and several global-scale observations products to constrain empirically the models projections. They show that the hydrological balance determines the Amazon forest's resilience to climate change, with a breakpoint at around 2000mm per year.

This study gives original and interesting insights on ESMs evaluation in the Amazon, trying to target the main actual source of uncertainty and to identify potential drivers of change. If I agree on the importance of the eco-hydrologic constrain to understand the Amazon response to climate, I have some serious concerns about the robustness of their conclusions, mainly because:

1/ The authors don't assess the uncertainties in the global validation products. For example, upscaled FLUXNET data products reasonably capture the seasonality and the mean annual values of evapotranspiration and GPP, however they underestimate extreme events from 30 to 60% at global scale (Jung et al., 2011). Fig.3a show a very different evapotranspiration response to precipitation, in terms of sensitivity response and threshold location, depending which dataset is used (MTE vs. MODIS MOS16). I think that fig.3 raises a caution flag about the conclusions made in this paper regarding uncertainties in global products.

2/ The paper relies on modeling results without taking in account missing mechanisms in LSMs: it is known that land surface model struggle to capture the Amazon forest productivity and biomass dynamics even when forced by local observed climate data (e.g. Joetzjer et al., 2015). Besides, fire dynamics, vegetation response to drought and tree mortality are very poorly represented (McDowell et al., 2011, Powell et al., 2013). For instance, Koven et al. 2015 showed that the carbon storage change in LSMs is dominated by NPP input changes whereas models do not produce any change in biomass turnover (mortality) except the HadGEM LSM model where NPP is arbitrarily prescribed to saturate irrespective of climate and CO2 driven changes in GPP when a grid cell is entirely covered by tropical evergreen vegetation. By contrast, several observational studies (e.g. Phillips et al., 2010, Lewis et al., 2011) point out to climate-induced changes in tree mortality across the Amazon during recent drought years. Thus, the missing representation of processes related to tree competition and mortality in LSMs makes the conclusion of the study by

Ahlstrom et al. very uncertain regarding future changes in the Amazon forest.

While the authors acknowledge the existence of these missing or badly represented processes, they didn't really state the potential implications of such well-known issue in their analysis. It is confusing considering the importance of such processes and their dependency (including feedback) on the hydrological cycle.

Specific comments

L53-54 Rephrase perhaps, uncertainties can emerge from both the terrestrial ecosystem model (e.g. poor representation of tree mortality) and climate bias. To me, this sentence suggests it's either one or the other.

L64 typo in "relationships"

L135 "... mortality and fire are poorly captured in most ESMs": give refs (for ESMs; eg. MC Dowell et al. 2011 TREE)

L309 ref

L456 Correct the reference (Schneider, U. et al. in FD_M_V6_050 (2011))

Extended figure 4-6, legend colors/models

Reviewer #1 (Remarks to the Author):

The manuscript the relationship between long-term GPP, ET and AGB to long-term precipitation (P), using an ensemble of nine CMIP5 ESMs, and a couple of observed datasets. They found that the above quantities have a linear relationship to P if $P < 2000$ mm/yr, being stable (or radiation-limited) if $P > 2000$ mm/yr. The authors go on and investigate these relationships using observed data, but filtering for different levels of land use, and conclude that these relationships become steeper if land use is higher. Their final conclusion is that reducing land use change emerges as a key opportunity to reduce vulnerability of Amazonian ecosystems.

I overall agree with the authors reasoning and conclusions. I just think their results are not new, and we have known this since the 2000s, may be since the 1990s. All these arguments have been repeatedly used by authors that discussed the Amazon tipping point. For example, the argument that deforestation would decrease the resilience of the Amazon rainforest was first presented by Shukla et al. (Science, 1990).

The main contribution of the authors may be to demonstrate that the relationships GPP, ET, AGB hold for the ensemble of the nine CMIP5 models they picked. However, the lack of predictability for future scenarios (“inconclusive support for such a transition”) makes the results less interesting. Again, even the inconclusiveness is not new, having first been reported by the first edition of C4MIP in 2006.

The way data is presented in Figure 4 is also new and interesting, but again, there is not any new information in it. It is well known that deforestation (increased land use) decreases AGB (there is nothing more obvious than that), ET, and GPP, and in the way they plotted their figures, these changes happen in the range of precipitation where land

use occurs.

My recommendation to the authors is to publish the analyses around Figures 1-3 in a more specialized journal, with a remark, however. The authors used only global datasets in their study, having cropped the Amazon region for their analyses. Global datasets can have significant regional biases, and uncertainties in empirical, or observed datasets, just cannot be avoided. I recommend using more regional studies in their analyses (they have cited very few regional studies in the specialized literature), and using an ensemble of observations in addition to their ensemble of models.

Reply: We thank the reviewer for acknowledging our contribution and suggesting what we should highlight and change. What we found is a strong eco-hydrological constraint on most processes and states in the Amazon forest and its transition to semi-arid savanna. While we agree that these type of relationships have been found and discussed individually in previous studies, we emphasize here the similarity in the shape and P breakpoint between these different processes and states (ET, GPP, AGB, tree cover) and the similarities between models and empirical datasets. We further discuss how these relationships can inform us on future responses to climate change and what key processes are involved.

In contrast to e.g. Shukla et al. 1990 that focused on feedbacks after land conversion, we assess drivers that can lead to a biome shift or major shift in functioning. Indeed, we do find that there is inconclusive support for a decrease in precipitation in the future, but we do find agreement among the ESMs on the response to changes in climate. In contrast to e.g. C4MIP and other studies that look at effects of CO₂ and climate on the land sink, we thereby clearly show what causes differences between models (precipitation biases) and what key processes are likely to govern future changes and potential die-back, i.e. a shift in the breakpoint, changes in precipitation and land use change. Further, to our knowledge, this is the first paper that shows that global ESMs essentially capture the main functioning of Amazonia (the ecosystem response to precipitation) and that mean and spatial precipitation bias are the main causes of disagreement between models and between models and data. By removing this veil of uncertainty and highlighting the key mechanisms we hope that new studies on the future of the Amazon rainforest and other tropical forests will benefit.

Our analysis also contributes to the discussion on bi-modality and ecosystem tipping points. As noted by the reviewer our analysis where we increasingly remove land use suggests that there is no marked general tipping point in natural ecosystems but that the majority of those patterns are caused by land use.

We agree that there are uncertainties in datasets. To address this we have added another dataset on ET and new analysis on differences between the datasets. We have performed a trend break analysis on the n=6 empirical datasets. We further discuss uncertainties and our reasoning on those uncertainties. In the conclusion we have clarified the contribution of the paper (see also responses to reviewer 3). Our analysis relies on gridded data which is the reason for the use of global gridded products that can be compared with global Earth system models. These datasets use available observations and satellite information in combination

with global patterns that help constrain their estimates and empirical models or machine learning algorithms. We are not aware of any regional gridded products with significantly lower uncertainties.

Changes: Update to Figure 3. New extended data figure 2 and 3. Update to Figure 4.

We made multiple changes throughout the manuscript, including changes to the text in the forest-savanna transition section to describe the new trend break analysis.:

Lines 117-120 now reads: *“The average breakpoint determined using a trend break analysis of the empirical datasets dependency on P is 2023 mm yr⁻¹ with a standard deviation of 128 mm yr⁻¹ (dashed red lines in Figure 3, breakpoint tree cover: 2075 mm yr⁻¹, Extended Data Figure 2).”*

Changes to the discussion includes a new section on uncertainties in datasets (lines 233-244), and a section on mortality and disturbances (lines 245-257). We also revised the last paragraph of the discussion, the conclusion (lines 287-295):

“Our results reconcile models and data and highlight the dominant role of the eco-hydrologic constraint in Amazonia. We found that climate biases, especially in precipitation, limit the current ability of individual models to represent the Amazon and likely other tropical forests accurately. Interactions between precipitation biases and land use are another cause of discrepancies among models and between models and empirical datasets. Although future CO₂ effects on WUE will likely increase GPP, increased climate variability can reduce carbon accumulation in Amazonia. The hydrological regime and its relationship to the breakpoint identified here may ultimately govern Amazon biomass and vegetation structure of native ecosystems.”

Reviewer #2 (Remarks to the Author):

One major set of related comments:

Line 160-169 – The reasoning here is very unclear. What does this sentence even mean: “Such bi-modality that overlaps on the P-axis does not necessarily cause a tipping point”? This passage is very poorly written.

Additionally, the reasoning overreaches substantially. It is quite clear and well known that the cerrado and other Amazonian grassy systems are used much more intensively than Amazonian forests. Thus, land use differs between forest and savannas. However, land use does not cause the cerrado to be where it is – after all, the cerrado clearly pre-dates colonial land use practices. So the directionality of the causality that is inferred here is totally wrong.

Response: Thank you for pointing out that this section was not clearly written. We have revised this section to clarify our statement on bi-modality and to clarify that our analysis does not suggest that the Cerrado is caused by land use. In the analysis of land use we looked at the deviation from the hydrological constraint, i.e. the amount of biomass or other variables that would likely be realized if there were no land use, but it was not our intention to suggest that the Cerrado is caused by land use nor does the analysis suggest that.

The section on lines 161-174 now reads:

“Similar effects of land use were found on ET, GPP and tree cover¹⁹ (Figure 4b-e). Although ET and GPP change less in heavily managed locations than AGB and tree cover do, all empirical datasets suggest that the mean response to P and the transition between ecosystem states is smooth when removing land use effects. The shift from a sharp decline to a smooth transition between ecosystem states when accounting for land use suggests that there is little risk for a general ecosystem state tipping point. Instead, the sharp transition is mainly caused by human transformation of highly productive ecosystems close to the breakpoint as well as widespread land use in systems with ~1500 mm annual P, which decrease the productivity and biomass in the semi-arid Cerrado. Our analysis does not exclude the existence of reported fire driven bi-modality⁵ which leads to separate and stable vegetation states⁴. Such bi-modality, with biomass at two distinct states for similar P does not necessarily cause a tipping point in our analysis of functional relationships which represent the average biomass in a given P interval. Overall, however, the deviation from the hydrological constraint observed here appears to be caused primarily by land use.”

Line 260-262 – This is a totally unsupported conclusion. Moreover, it’s not even necessary. The other key conclusions are worthwhile and interesting. It’s also quite clear that, today, fires within forest have anthropogenic origins related to land use change – so you can make the same exact conclusion on line 264 without being unnecessarily combative about fire-vegetation feedbacks.

So, if you remove Lines 160-169 and Lines 260-262, the paper is interesting and makes real and worthwhile contributions to the literature on Amazon forest susceptibility to global change.

Reply: We removed the statement on natural fires on lines 160-169 (see answer to comment above) but kept and clarified some of the discussion on bi-modality and the tipping point to explain how these concepts differ. We further followed the reviewer's advice and revised concluding paragraph.

The conclusion now reads (lines 287-295):

“Our results reconcile models and data and highlight the dominant role of the eco-hydrologic constraint in Amazonia. We found that climate biases, especially in precipitation, limit the current ability of individual models to represent the Amazon and likely other tropical forests accurately. Interactions between precipitation biases and land use are another cause of discrepancies among models and between models and empirical datasets. Although future CO₂ effects on WUE will likely increase GPP, increased climate variability can reduce carbon accumulation in Amazonia. The hydrological regime and its relationship to the breakpoint identified here may ultimately govern Amazon biomass and vegetation structure of native ecosystems.”

Some minor comments:

You should introduce fire more completely in the introduction, given how fundamental it is to your later analyses. You could also or alternatively reframe a bit to make fire a bit less central, which you should probably do anyways, as described above.

Reply: We have followed your advice and made fires less central in the conclusions because, as the reviewer points out, it is not in the main focus of the analysis. Please see answers above and changes throughout the paper.

Line 44 – P = phosphorus, and abbreviating like this in the introduction is unnecessary/confusing.

Reply: We now clarify that P is short for precipitation in the introduction.

Line 97 – should be ‘relationships’

Reply: Thanks, changed.

Line 179 – most geographers would describe the central Amazon as being in the region of Manaus. There are no semi-arid savannas anywhere near there. Do you mean central South America? Central Brazil? And actually, a good part of your study region is not actually in Amazonia.

Reply: Thanks. We made a distinction between “Amazon-basin” and “Amazonia” where the latter includes also surrounding areas. To clarify this we have added an explanation at first mentioning of Amazonia.

Lines 49-50: “*(here defined as the Amazon basin and surrounding semi-arid vegetation)*”

Reviewer #3 (Remarks to the Author):

General comments

In this study, the authors investigate the problem of the Amazon forest vulnerability to climate change by using a subset of CMIP5 earth system models (ESM) and several global-scale observations products to constrain empirically the models projections. They show that the hydrological balance determines the Amazon forest's resilience to climate change, with a breakpoint at around 2000mm per year.

This study gives original and interesting insights on ESMs evaluation in the Amazon, trying to target the main actual source of uncertainty and to identify potential drivers of change. If I agree on the importance of the eco-hydrologic constrain to understand the Amazon response to climate, I have some serious concerns about the robustness of their conclusions, mainly because:

1/ The authors don't assess the uncertainties in the global validation products. For example, upscaled FLUXNET data products reasonably capture the seasonality and the mean annual values of evapotranspiration and GPP, however they underestimate extreme events from 30 to 60% at global scale (Jung et al., 2011). Fig.3a show a very different evapotranspiration response to precipitation, in terms of sensitivity response and threshold location, depending which dataset is used (MTE vs. MODIS MOS16). I think that fig.3 raises a caution flag about the conclusions made in this paper regarding uncertainties in global products.

Reply: We have added an additional ET datasets (GLEAM v3.0a) to better illustrate uncertainties in ET stemming from empirical datasets. We further performed a trend break analysis on all empirical datasets to better visualize the breakpoints in the datasets. The trend break analysis showed a mean breakpoint at 2023 mm across the six datasets, with a range in ET from 1826-2226 mm, which is slightly larger than found in the separate study by Guan et al that estimate the range to be between 1850 and 2150 mm (described on lines 117-120 and in new Extended Data Figure 2). Overall, we hope that the general consistency between variables and datasets strengthens our argumentation that the breakpoint is somewhere around 2000 mm while better illustrating uncertainties in ET and empirical datasets.

We further added a section in the discussion on dataset uncertainties. Here we describe the major cause of uncertainties and our approach to account for uncertainties.

Lines 233-244:

“The empirical datasets used here have their own uncertainties. Uncertainties in individual datasets are difficult to assess because they originate in part from sparse observations (e.g. flux towers³⁵ or weather stations^{36,37}) and from clouds interfering with satellite observations in the wet season³⁸. Our approach to address the uncertainty includes using multiple independent empirical datasets that represent both ecosystem fluxes and states. All datasets

show a similar breakpoint between water and radiation limitation (Extended Data Figure 2). This breakpoint apparent from the functional relationships between different ecosystem states and fluxes and precipitation is also similar to the breakpoint found previously using interannual variations in ecosystem productivity and a gravity informed water balance model²¹. Together the coherence in the existence and location of the breakpoint across ET, GPP, AGB and tree cover and previous work lends support to the existence of a breakpoint with a value close to 2000 mm annual P.”

2/ The paper relies on modeling results without taking in account missing mechanisms in LSMs: it is known that land surface model struggle to capture the Amazon forest productivity and biomass dynamics even when forced by local observed climate data (e.g. et al., 2015). Besides, fire dynamics, vegetation response to drought and tree mortality are very poorly represented (McDowell et al., 2011, Powell et al., 2013). For instance, Koven et al. 2015 showed that the carbon storage change in LSMs is dominated by NPP input changes whereas models do not produce any change in biomass turnover (mortality) except the HadGEM LSM model where NPP is arbitrarily prescribed to saturate irrespective of climate and CO2 driven changes in GPP when a grid cell is entirely covered by tropical evergreen vegetation. By contrast, several observational studies (e.g. Phillips et al., 2010, Lewis et al., 2011) point out to climate-induced changes in tree mortality across the Amazon during recent drought years. Thus, the missing representation of processes related to tree competition and mortality in LSMs makes the conclusion of the study by Ahlstrom et al. very uncertain regarding future changes in the Amazon forest.

While the authors acknowledge the existence of theses missing or badly represented processed, they didn't really state the potential implications of such well-known issue in their analysis. It is confusing considering the importance of such processes and their dependency (including feedback) on the hydrological cycle.

Reply: We thank the reviewer for this important comment. Our approach relies on functional relationships and does not directly analyze the impact of changing climate over a given ecosystem, as e.g. a timeseries of regionally averaged biomass from a future climate change scenario simulation would do. That said, we do agree that the potential for a different response to future novel climates remains unresolved. To address this in the paper we have added a section that discuss the models (in)abilities to simulate mortality and how that could affect the functional relationships. We hope that our analysis can clarify and highlight these uncertainties by explaining the main causes for present day model divergence (climate-biases).

New section on lines 245-257:

“Our assessment of future changes in Amazonian ecosystems is described as a response to P but does not assess the mean ecosystem response to changing climate. We thereby control for the large uncertainty between simulated changes in climate, especially P (Extended Data Figure 9), which is a large cause for reported divergent future predictions of Amazonian

ecosystem fluxes and structure^{39,40}. However, the ability of ESMs to accurately simulate Amazon responses to future climate may be compromised by relatively poor representation of drought mortality and disturbances,^{25,41} leading to large uncertainties in vegetation turnover⁴² and resulting biomass. Uncertainties in the models' ability to capture the occurrence and the ecosystem response to potentially more extreme weather and hotter droughts may also affect the future responses to P presented here. This potential impact of increased drought mortality remains to be evaluated and requires a new generation of ESMs with land models that include detailed representation of vegetation dynamics, disturbances and mortality, with accurate predictions of future climate and weather extremes.”

Specific comments

L53-54 Reword perhaps, uncertainties can emerge from both the terrestrial ecosystem model (e.g. poorly representation of tree mortality) and climate bias. To me, this sentence suggests it's either one or the other.

Reply: Thanks, this section now reads (lines 53-56): *“This uncertainty originates from several issues, mainly terrestrial ecosystem sub-models, biases in climate generated internally in the ESMs, or both. The effect of climate biases on uncoupled carbon cycle simulations has been shown to have a large impact on Amazonia¹³.”*

L64 typo in “relationships”

Reply: Fixed.

L135 “... mortality and fire are poorly capture in most ESMs”: give refs (for ESMs; eg. MC Dowell et al. 2011 TREE)

Reply: Thanks.

L309 ref

Reply: Fixed.

L456 Correct the reference (Schneider, U. et al. in FD_M_V6_050 (2011))

Reply: Thanks, fixed.

Extended figure 4-6, legend colors/models

Reply: Thanks, fixed.

Reviewers' Comments:

Reviewer #1:

Remarks to the Author:

The authors have satisfactorily responded my comments, including on the novelty of their work with respect to the previous literature. They have also included new analyses based on additional observed datasets, which strengthened the manuscript. I recommend publication.

Reviewer #2:

Remarks to the Author:

On the whole, I found the manuscript interesting, engaging and informative. I suggest only a few minor changes below (only one of which is philosophically important, but even so, not much work).

Line 30: I think most people prefer an extra comma in situations such as these, e.g. "A, B, and C converge...". (same in the next sentence, "A, B, and C align on...")

Line 173-174: I commend you for toning down the language on the idea of a fire-driven tipping-point. However, I can see here that you are not at all convinced by the idea that land use may not be the driver, but may instead be the result of savanna-type vegetation structure. I would suggest toning down this strong (if more gently worded) conclusion here – as before, it is not particularly well-supported by a more expansive interpretation of your analyses.

Again, I don't see this is as a major (or any kind of blow) to your major conclusions. That land use impacts ABG, we know, and this can totally swamp all climate effects on ABG. Your major point is that a dry bias over the central Amazon creates the illusion that the Amazon is at risk of crossing a tipping point, while in fact that tipping point, if it exists, is certainly not located over the central Amazon. Your analysis demonstrates this point nicely, and the existence of stable savannas elsewhere does nothing to change that conclusion at all. I agree (despite my objections above) that the risk of Amazon collapse may have been quite overrated in the literature.

I would simply suggest a change in the last sentence to read: "Overall, however, the deviation from the hydrological constraint observed here appears, currently, to be related primarily to land use." No other change is necessary.

Reviewer #3:

Remarks to the Author:

The two main concerns I had on the manuscript were well taken into account by the authors in the revised version. Increasing the number of independent datasets of evapotranspiration strengthens the conclusion on the quantification of the breakpoint and illustrates the uncertainties of the global datasets. I also appreciated the discussion about the missing processes in DGVM. I think the revised version is suitable for publication.

Reviewer #1 (Remarks to the Author):

The authors have satisfactorily responded my comments, including on the novelty of their work with respect to the previous literature. They have also included new analyses based on additional observed datasets, which strengthened the manuscript. I recommend publication.

Reply: We thank the reviewer for acknowledging our changes.

Reviewer #2 (Remarks to the Author):

On the whole, I found the manuscript interesting, engaging and informative. I suggest only a few minor changes below (only one of which is philosophically important, but even so, not much work).

Reply: We thank the reviewer for acknowledging our study and our changes.

Line 30: I think most people prefer an extra comma in situations such as these, e.g. “A, B, and C converge...”. (same in the next sentence, “A, B, and C align on...”)

Reply: Thanks, changed.

Line 173-174: I commend you for toning down the language on the idea of a fire-driven tipping-point. However, I can see here that you are not at all convinced by the idea that land use may not be the driver, but may instead be the result of savanna-type vegetation structure. I would suggest toning down this strong (if more gently worded) conclusion here – as before, it is not particularly well-supported by a more expansive interpretation of your analyses.

Again, I don't see this is as a major (or any kind of blow) to your major conclusions. That land use impacts ABG, we know, and this can totally swamp all climate effects on ABG. Your major point is that a dry bias over the central Amazon creates the illusion that the Amazon is at risk of crossing a tipping point, while in fact that tipping point, if it exists, is certainly not located over the central Amazon. Your analysis demonstrates this point nicely, and the existence of stable savannas elsewhere does nothing to change that conclusion at all. I agree (despite my objections above) that the risk of Amazon collapse may have been quite overrated in the literature.

I would simply suggest a change in the last sentence to read: “Overall, however, the deviation from the hydrological constraint observed here appears, currently, to be related primarily to land use.” No other change is necessary.

Reply: We thank the reviewer and agree, the sentence has been changed as suggested.

Reviewer #3 (Remarks to the Author):

The two main concerns I had on the manuscript were well taken into account by the authors in the revised version. Increasing the number of independent datasets of evapotranspiration strengthens the conclusion on the quantification of the breakpoint and illustrates the uncertainties of the global datasets. I also appreciated the discussion about the missing processes in DGVM. I think the revised version is suitable for publication.

Reply: We thank the reviewer for acknowledging our changes.